# A Review Discussing Synthesis and Translational Studies of Medicinal Agents Targeting Sphingolipid Pathways

**DOI:** 10.3390/biom15071022

**Published:** 2025-07-16

**Authors:** Sameena Mateen, Jordan Oman, Soha Haniyyah, Kavita Sharma, Ali Aghazadeh-Habashi, Srinath Pashikanti

**Affiliations:** Biomedical and Pharmaceutical Sciences Department, Kasiska Division of Health Sciences, L.S. Skaggs College of Pharmacy, Idaho State University, Pocatello, ID 83209-8288, USA; jordanoman@isu.edu (J.O.); sohahaniyyah@isu.edu (S.H.); kavitasharma@isu.edu (K.S.); aliaghazadehhabas@isu.edu (A.A.-H.)

**Keywords:** sphingolipids, natural products, ceramide, chiral pool, enantioselective synthesis

## Abstract

Sphingolipids (SLs) are a class of bioactive lipids characterized by sphingoid bases (SBs) as their backbone structure. These molecules exhibit distinct cellular functions, including cell growth, apoptosis, senescence, migration, and inflammatory responses, by interacting with esterases, amidases, kinases, phosphatases, and membrane receptors. These interactions result in a highly interconnected network of enzymes and pathways, known as the sphingolipidome. Dysregulation within this network is implicated in the onset and progression of cardiovascular diseases, metabolic disorders, neurodegenerative disorders, autoimmune diseases, and various cancers. This review highlights the pharmacologically significant sphingoid-based medicinal agents in preclinical and clinical studies. These include myriocin, fingolimod, fenretinide, safingol, spisulosine (ES-285), jaspine B, D-e-MAPP, B13, and α-galactosylceramide. It covers enantioselective syntheses, drug development efforts, and advances in molecular modeling to facilitate an understanding of the binding interactions of these compounds with their biological targets. This review provides a comprehensive evaluation of chiral pool synthetic strategies, translational studies, and the pharmacological relevance of sphingolipid-based drug candidates, offering a pathway for future research in sphingolipid-based therapeutic development.

## 1. Introduction

The term sphingolipids (SLs) is derived from the Greek word “sphinx” owing to their enigmatic biochemical properties [1,2]. Sphingolipids (SLs) form integral components of functional lipid rafts in eukaryotic cells. These microdomains are composed of organized SLs, along with transmembrane, structural, and functional proteins [3,4]. They play a significant role in membrane biochemistry and in regulating cellular function [5]. The chemical structure of sphingolipids constitutes a sphingoid base, such as sphingosine (Sph), as the building block. Sph is a substrate for the biosynthesis of diverse complex sphingolipids (Figure 1). These complex sphingolipids are involved in highly regulated metabolic processes. Despite SLs’ structural and functional diversity, they are created and recycled by common catabolic biosynthetic pathways [3,4,5]. These biosynthetic pathways occur in different organelles, forming an array of interconnected networks that differ from a single common entry point and converge into a single breakdown pathway [6]. This network of SLs, known as the sphingolipidome, is in proximity to various receptors, enzymes, and lipids involved in critical cell functions, including growth regulation, apoptosis, senescence, cell migration, adhesion, and inflammatory responses. Most of these pharmacological functions have implications in diseased states, such as cancer, cardiovascular disorders, metabolic disorders, and immune function [6]. The spatial distribution of sphingolipids in distinct compartments has evolved to serve specialized functions in eukaryotic cells [7].

The physicochemical properties of SLs constitute a sphingoid base with an amine-alcohol functionality with an inherent *L*-stereochemistry [4,8]. Biosynthesis begins with the cellular chiral pool, _L_-Ser, resulting in a family of oxysphingolipids, while _L_-Ala results in a family of deoxysphingolipids [9] (Figure 1). The SL biosynthesized from _L_-Ala lacks primary alcohol functionality, which is an essential nucleophile for derivatization towards complex functional SLs. For example, ceramide (Cer) is biosynthesized from _L_-Ser; the primary alcohol of Cer is derivatized to afford a very important sphingolipid, sphingomyelin (SM). SM is a basic unit of the myelin sheath, among several others. However, a deoxy-ceramide cannot be derivatized as SM due to a lack of alcohol functionality. Deoxysphingolipids also differ in their physicochemical properties due to the absence of the primary alcohol [10].

SLs possess the characteristic ‘sphingoid-base’, an *anti*-amino alcohol with varying aliphatic chains [11] (Figure 1). The *anti*-amino alcohol system imparts a polar nature, with both amine and alcohol functionalities having a lone pair of electrons. The ‘-OH’ is the H-bond donor, whereas the -NH_2_ can participate both as an H-bond donor and acceptor, and the basic nature of -NH_2_ makes this core the “Sphingoid base”. Amine, as a base, can participate as a biological nucleophile. The *anti*-amino alcohol system exhibits a stable conformation and is enantiopure, allowing for derivatization towards complex functional sphingolipids. Synthetically, achieving a 100% diastereoselective *anti*-amino alcohol is very challenging. However, during the biosynthesis of Cer, an elegant catalysis by 3-keto-sphinganine reductase affords this selectivity (Figure 2). A gauche or eclipsed confirmation along the σ-bond would have experienced an internal H-bond and interfered with further derivatization towards functional SLs. So, *anti*-amino alcohol is favored.

## 2. Ceramide Biosynthesis and Metabolism

Ceramide is a biochiral molecule and a key sphingolipid, serving as the central hub within the sphingolipidome network [12]. This intricate web with ceramide as the central signaling molecule encompasses various sphingolipid molecules and their metabolic pathways, including biosynthesis, breakdown, and modification of different sphingolipid species. Ceramide also regulates the cross-talk between multiple cellular metabolic processes, including, but not limited to, apoptosis, senescence, cell cycle arrest, cell aging, and various responses to stress signals within the body [13,14].

Cer biosynthesis involves *N*-acylation of the sphingoid bases with varying fatty acids and/or a variety of head group substitutions to afford complex SLs. The aliphatic tail can differ in structure depending on the number of carbons in the chain, the substitution of unsaturation, the position of these double bonds, and even the spatial arrangement. Cer biosynthesis involves complex pathways catalyzed by specific enzymes. There are three main pathways involved in ceramide biosynthesis: the *de novo* biosynthesis pathway, which starts from _L_-Serine; the sphingomyelinase (SMase) pathway, which involves the hydrolysis of sphingomyelin; and the salvage pathway, which hydrolyzes complex sphingolipids. These three pathways contribute to the chiral pool ceramide cellular levels.

The significance of spatial distributions of enzymes in different cellular compartments and their effects on cell signaling and SL flux was identified early on by Hannun et al. [15]. For example, the rate-limiting enzyme of the *de novo* pathway, *serine palmitoyl transferase* (SPT), is located in the endoplasmic reticulum [16]. Dihydroceramide (DhCer), a substrate for the Cer biosynthesis in the *de novo* pathway, is biosynthesized from the endoplasmic reticulum-derived dihydrosphingosine. Meanwhile, Sph, for Cer regeneration via the salvage pathway, is generated from the breakdown of complex SLs in lysosomes [17]. The distinct ceramide/dihydroceramide synthase isoforms involved in the *de novo* versus salvage pathways may be of differential cellular localization [15]. Therefore, the ceramide biosynthetic pathways and the sphingolipid metabolism are highly compartmentalized with cross-talk with complex networking involving transfer protein [18].

### 2.1. De Novo Biosynthesis Pathway

The *de novo* pathway builds up a cellular ceramide pool to meet the demand of rapidly dividing cells by utilizing a simple starting material, _L_-ser [19]. The initial catalysis is localized to the cytosolic face of the endoplasmic reticulum, differentiating the polarity. The first step is catalyzed by serine palmitoyl transferase (SPT). It is the rate-determining step (RDS) involving a Claisen-type condensation of _L_-Ser and the palmitoyl-CoA utilizing co-factor PLP (pyridoxal 5′-phosphate) to afford 3-ketosphinganine (Figure 3).

Several medicinal chemistry approaches targeting SPT were employed to leverage the RDS and control the accumulation of cellular ceramide.

An elegant mechanistic model from Yard, B.A., et al. [20] depicts the formation of internal aldimine (Figure 3) with PLP (green) covalently bound to Lys-265 (blue) in the enzyme’s active site, with the substrate Ser. The pentadentate phosphate anion of PLP is near the carboxylate of Ser-264. The resulting β-ketone (Ketosphinganine) undergoes alkyl chain modifications catalyzed by 3-ketodihydrosphingosine reductase, ceramide synthases, and finally dihydroceramide desaturase, resulting in the formation of chiral pool ceramide (Figure 2). Activation of this *de novo* biosynthetic pathway towards ceramide flux is observed in several abnormalities [21]. For example, during apoptosis, signaling occurs in response to chemotherapeutic agents [22], cannabinoids [23,24], and stress reactions [19].

### 2.2. Sphingomyelinase (SMase) Pathway

The SMase pathway involves the hydrolysis of membrane-embedded sphingomyelin by the action of a group of enzymes, sphingomyelinases (SMases), resulting in Cer and charged phosphorylcholine [25]. The sphingomyelinase enzyme is primarily found in the brain and lysosomes. Mutations in the gene encoding sphingomyelinase result in Niemann–Pick type A and type B, which are lysosomal storage diseases characterized by neuropathic and fatal outcomes [26]. SMases can be either acidic or neutral, and the activity of these enzymes can be significantly stimulated by exposure to stress signals, such as TNF-alpha, oxidative stress, or Fas ligand [27]. SMases play a vital role in ceramide-related signal transduction. In mammals, four different types of SMases have been identified, including SMase1 (SMPD2), SMase2 (SMPD3), SMase3 (SMPD4), and a mitochondrial-associated SMase (SMPD5) [28]. The first mammalian SMase to be cloned was SMPD2 [29]. SMPD2 is primarily localized to the endoplasmic reticulum and nuclear matrix, requiring divalent cations (Mg^2+^) for activation [28]. The metal-based catalysis suggests a possible lone pair of electrons coordinating with the metal Mg. Another related metalloenzyme, bacterial phospholipase C (PLC), catalyzes the hydrolysis of the phosphate attached to the glycerol of the sphingomyelin polar head group [30].

### 2.3. The Salvage Pathway

The “salvage pathway” is also known as the “sphingolipid recycling” pathway because it is a metabolic process where cells effectively recycle the components of broken-down sphingolipids [5]. The complex sphingolipids are broken down into sphinganine (SA), the basic sphingolipid, which is then used to synthesize Cer. Various key enzymes are involved in this process, like the SMases, glucocerebrosidase (acid-β-glucosidase), dihydroceramide synthases, and ceramidases (Figure 2). This process occurs in the acidic subcellular compartments, lysosomes, and late endosomes. This salvage pathway leading to the regeneration of Cer contributes to nearly 50–90% of sphingolipid biosynthesis [31].

## 3. Biological Significance of Sphingolipids and Derivatized Sphingolipids

Bioactive sphingolipids have emerged as critical molecules in various disease states; this is emphasized by approximately 30,000 studies published over the past two decades, as shown in a PubMed search (Figure 4). This growing body of literature reflects the continuous and increasing interest in sphingolipid metabolism and its therapeutic potential.

The main sphingolipids contributing to biological relevance are the complex sphingolipids (SLs) obtained through derivatization. The major derivatized bioactive sphingolipids consist of ceramide (Cer), glucosylceramide (GCer), ceramide-1-phosphate (C1P), sphingosine (Sph), sphinganine (SA), and sphingosine-1-phosphate (S1P) as depicted in Figure 5. These are derivatives of the *anti*-amino alcohol system in the sphingoid base. The specific biological processes in which these derivatized functional sphingolipids are involved include cell migration, proliferation, inflammation, and apoptosis [3,32]. These cellular signaling pathways are relevant to autoimmune disorders, cardiovascular diseases, infectious diseases, inflammation, lysosomal storage diseases, and cancer.

Disrupted sphingolipid metabolism across various human cancers suggests that bioactive SLs are critical in supporting tumor development, maintaining cancer cell viability, and regulating tumor cell death and survival [33]. The rapid division and growth of cells require membranous organelles for their development. Thus, the best cancer treatments and therapeutics can be developed by carefully controlling the flux of sphingolipids by perturbation of Cer/S1P rheostat [34]. The therapeutic roles of sphingolipids in various diseases, most notably in cancer, have been elucidated significantly in the past few decades, as shown in Table 1. Very few potential drug candidates have been identified, and the full potential of bioactive lipids is yet to be fully realized in the context of ceramide flux and lipid raft function.

## 4. Sphingolipid Rheostat and Its Significance in Cancer

Research efforts from the Hannun, Obeid, Lynch, Ogretmen, and Santos groups have extensively contributed to advancements in sphingolipid biochemistry and drug discovery efforts [3,6,45]. Cer is considered pro-apoptotic, contributing to various cell death pathways that involve cellular functions such as growth arrest, differentiation, and programed cell death [3,6,46]. Sphingosine (Sph), which is produced by the breakdown of Cer, is also pro-apoptotic, contributing to cell death [47]. S1P is a pro-survival SL that promotes cell growth and proliferation [48], the opposite of Cer functioning. The evolution of sphingolipid biochemistry with these two factors of opposing biological relevance is prominent in determining the fate of the cell, leading to either cell survival or cell death (Figure 6).

The perturbation of this axis is observed in Cer’s apoptotic effects [49]; elevated Cer levels, resulting from cellular stress, trigger apoptosis and cell death. In response to high ceramide (Cer) levels, enzymes such as sphingosine kinase (Sphk) become activated, producing the anti-apoptotic sphingosine-1-phosphate (S1P), which counteracts the pro-apoptotic effects of Cer [13]. Cer, a tumor-suppressing lipid, mediates the pro-apoptotic processes, such as growth arrest, senescence, differentiation, and apoptosis. Conversely, the tumor-promoting lipid S1P promotes cell proliferation, transformation, migration, inflammation, and angiogenesis [50]. The individual Cer, Sph, or S1P levels do not determine the cell’s fate; it is determined by the ratio between the Cer + Sph and the S1P levels, which is often referred to as the sphingolipid rheostat or the “Cer/S1P rheostat” [51,52].

Using medicinal chemistry approaches, the sphingolipid rheostat can be modulated to control the cell’s fate, leading to either cell survival or cell death. These probes have the potential to identify hits targeting several disease states, including neurodegenerative disorders, inflammatory conditions, and, most importantly, various types of cancer [34,53].

Shown in Table 2 are small molecules targeting various enzymes in the sphingolipid rheostat.

This review discusses the synthesis and biological significance of sphingoid-based small molecules, as well as their relevance to preclinical and clinical studies targeting enzymes involved in sphingolipid biochemistry. Due to space limitations, we have focused on selected natural products and small molecules that have reached preclinical and clinical stages. These include myriocin, fingolimod, fenretinide, safingol, spisulosine (ES-285), jaspine B, D-e-MAPP, B13, and α-galactosylceramide (the bolded compounds in Table 2). We acknowledge the tremendous efforts in medicinal chemistry made by other groups and apologize if we have overlooked their contributions [101,102,103].

## 5. Sphingolipid-Based Therapeutic Agents

These molecules, due to their structural similarities with ceramide, have demonstrated potential therapeutic efficacy in various disease conditions, particularly cancer. Most biochemical assays suggest modulation of the sphingolipid rheostat and its perturbation. In this review, we have discussed in detail the stereoselective, viable synthetic procedures performed by various medicinal chemists, as well as their biological significance, including preclinical and clinical studies available in the literature. The enzyme targets for these compounds are known (refer to bolded compounds in Table 2); we have added the enzyme structures downloaded from the RCSB protein database.

### 5.1. Myriocin

Myriocin, also known as ISP-1, as shown in Figure 7, is a biologically active sphingoid base with immunosuppressive properties. It was first isolated from the fungus *Isaria sinclairii* [104]. Myriocin is a potent inhibitor of serine-palmitoyl-transferase (SPT), which is the very first and the rate-determining enzyme of the *de novo* ceramide biosynthetic pathway [105]. Based on the SPT crystal structure (Figure 8), the active site involves the formation of a Schiff base between the substrate _L_-Ser and the cofactor PLP (PDB: 3A2B, Figure 8).

Myriocin’s physicochemical characteristics include a charged ammonium and carboxylate group in physiological conditions, with a lipophilic tail. Its biological relevance has encouraged several medicinal and synthetic efforts, contributing to early pharmaceutical development towards understanding immunological applications.

#### 5.1.1. Synthesis of Myriocin

Myriocin was first synthesized by Yoshikawa et al. [106] (Figure 1). Of the several total syntheses, discussed below is an application of the chiral pool strategy starting from a chiral synthon, 2-deoxy-D-glucose. The key reactions involve acetonide protection to afford intermediate **1.1**. Subsequent alcohol oxidation under Swern conditions resulted in a keto intermediate **1.2**, followed by base-mediated carbanion addition using Darzen’s reaction, which afforded **1.3**. Gem-dichloro intermediate (**1.3**) is subjected to multi-step synthesis involving protection–deprotection, S_N_^2^, the Wittig reaction, and regioselective separation to afford myriocin (Figure 1) [106].

Another total synthesis of myriocin by Nambu, Hisanori, et al., which involves a synthetic route that utilizes the Du Bois mechanism for the construction of a quaternary chiral center, is discussed in Figure 2. The key reactions involved are Rh (II)-catalyzed C-H amination of a sulfamate and olefin metathesis strategy. Reaction of a sulfamate with Ph(OAc)_2_ and MgO in the presence of Rh_2_(OAc)_4_ followed by vinyl addition afforded oxathiazinane *N*, *O*-cyclic sulphonate ester as a single product (**2.2**) with a quaternary center. The olefinic handle in **2.2** was subjected to cross-metathesis with a ketone-protected olefin in presence of Grubbs catalyst (gen II) to establish the required long olefinic side chain, on further deprotection and oxidation of primary alcohol group afforded the aldehyde, which on intramolecular lactonization afforded **2.3**. A final deprotection followed by lactone hydrolysis afforded enantiopure myriocin (Figure 2) [107].

#### 5.1.2. Biological Significance of Myriocin

Myriocin is an SPT inhibitor that suppresses T-cell proliferation by modulating sphingolipid metabolism. It is also a potent immunosuppressant, inhibiting the proliferation of an IL-2-dependent mouse cytotoxic T-cell line, CTLL-2, at nanomolar concentrations [108].

Myriocin inhibited cell growth in A549 and NCI-H460 lung cancer cell lines by inducing apoptosis through the activation of the p-JNK, p-p38, and DR4 pathways. It also synergistically enhanced the anti-cancer effects of notable drugs, such as docetaxel and cisplatin [108].

Several articles depict the crystal structure of serine palmitoyltransferase and myriocin bound in the active site of the enzyme; one of these figures is shown in Figure 9 [109].

The biological importance of myriocin has encouraged extensive SAR studies, which resulted in various analogs inspiring the synthesis of an amino alcohol-based scaffold. For example, VPC03090 and OSU-2S (Figure 10) have structural similarities with myriocin. These analogs have exhibited anticancer properties in in vitro and in vivo models. The preliminary studies identified their significance in S1P receptor antagonism in hepatoma, hepatocellular carcinoma, and gastric, prostate, and breast cancers [110]. The S1P receptor is a G-protein-coupled receptor that activates with the binding of S1P extracellularly.

### 5.2. Fingolimod

Fingolimod (Gilenya), as shown in Figure 11, is a sphingosine 1-phosphate receptor modulator, was approved by the FDA in 2010 for treating relapsing–remitting multiple sclerosis (MS) [111].

#### 5.2.1. Synthesis of Fingolimod

The key step in synthesizing fingolimod involved developing a hydrophilic ‘2-aminopropane-1,3-diol’ head group. An early synthesis of fingolimod utilized a base-mediated alkylation of diethyl 2-acetamidomalonate (**3.1**) with 1-(2-chloroethyl)-4-octyl benzene (**3.2**) to yield the Adachi–Fujita intermediate diethyl 2-acetamido-2-(4-octyl phenethyl)malonate (**3.3**). Complete reduction of the diethyl esters, followed by amide deprotection, provided fingolimod (Figure 3) [112].

The first reported synthetic scheme involved developing the polar head group first, followed by the lipophilic tail, which required extensive chromatography resulting in reduced yields. The unstable reagent (**3.2**) was another concern, as it affected the cost of the synthesis on a larger scale.

The improved synthesis of fingolimod included a late-stage incorporation of a polar head group, leading to a scalable and cost-effective synthesis by Kandagatla et al. [113]. (Figure 4). The synthesis involved the alkylation of diethyl 2-acetamidomalonate, followed by complete reduction with NaBH4, yielding the polar diol (**4.3**) and acetyl group protection leading to **4.4**. The lipophilic tail was introduced via Friedel–Crafts acylation on **4.4** to yield **4.5**, a keto intermediate. Subsequent keto transformation provides the required saturated lipophilic tail, **4.6**. Global deprotection with 6N HCl produces fingolimod as the HCl salt [113].

#### 5.2.2. Pharmacological Significance of Fingolimod

Fingolimod is an FDA-approved immunotherapy drug for relapsed multiple sclerosis with a novel mechanism of action. Fingolimod targets S1P as it resembles its ligand and rapidly phosphorylates into fingolimod-P, thus competing with it to bind to the G-protein-coupled S1P receptors (S1P_1,3,4,5_) [114,115]. It prevents lymphocyte migration by ceasing them in the lymph nodes. It also inhibits lymphocyte production in secondary lymphoid organs, decreasing the number of lymphocytes in systemic circulation and infiltration into the brain and finally attenuating the inflammation [116,117].

Fingolimod also inhibits the PI3K/Akt/mTOR/p70 S6K signaling pathway, thereby reducing the migration and invasion of human glioblastoma cell lines [118], the primary cause of death in glioblastoma patients [119]. It is also a cannabinoid receptor antagonist that inhibits cPLA2 [120] and ceramide synthase [121]. Fingolimod has also been shown to be a promising treatment option for patients infected with the SARS-CoV-2 virus, as it reduces lung inflammation and prevents pulmonary exudation [122,123].

### 5.3. Fenretinide

#### 5.3.1. Synthesis of Fenretinide

Fenretinide, *N*-4-Hydroxyphenyl-Retinamide (4HPR), as shown in Figure 12, is a synthetic derivative of all-trans-retinoic acid (ATRA) [124]. Structurally, fenretinide is a polyene conjugated system. Gander et al. from Johnson & Johnson completed the first synthesis of Fenretinide in 1960, to explore its potential as a dermatological agent [125]. However, unlike several retinoic acid analogs, it did not exhibit activity as a dermatological agent.

Discussed below is a single-step coupling reaction between all-trans-retinoic acid (**5.1**) and 4-aminophenol (**5.2**), affording fenretinide (Figure 5) [126]. 

#### 5.3.2. Pharmacological Significance of Fenretinide

It was not until the 1990s that fenretinide was investigated for its anticancer potential. Fenretinide primarily works against cancer cells by apoptosis through the accretion of ceramides and reactive oxygen species (ROS) within tumor cells. Fenretinide is one of the few retinoids that exhibits high in vitro cytotoxic activity against a wide range of targets and several cancer cell lines, including those of the lung, breast, ovarian, head and neck, prostate, and skin cancer cells [127,128,129,130,131]. Since then, it has been and continues to be evaluated in clinical trials in specific types of cancer, such as ovarian, prostate, neuroblastoma, cervical, lung, renal, bladder, breast, glioma, head and neck carcinoma, skin non-Hodgkin’s lymphoma, and Ewing’s sarcoma. Fenretinide can act in a retinoid receptor-dependent manner or independently of the nuclear retinoid receptor [132]. Fenretinide thus became a promising candidate for various cancer types, often by disrupting the normal lipid metabolism within cancer cells, leading to cell death, particularly in breast cancer, due to its selective accumulation in fatty tissue [133]. The clinical phase I-III evaluations of fenretinide have shown minimal systemic toxicity and tolerability [134,135]. However, fenretinide is not FDA-approved, as the main findings of all the clinical trials are not universally positive due to poor bioavailability and hydrophobicity. But IND status encouraged pharmaceutical development.

The pharmacological significance of fenretinide led to the development of novel formulations to overcome the low bioavailability. The notable ones are the complexation of fenretinide with 2-hydroxypropyl-beta-cyclodextrin, which resulted in nanoformulations and nano-fenretinide that exhibited increased bioavailability and therapeutic effectiveness [135]. A micellar formulation—bionanofenretinide—showed enhanced bioavailability, low toxicity, and potent antitumor efficacy in lung, colorectal cancers, and melanoma xenografts [136]. An emulsion intravenous infusion yielded better results than earlier capsule formulations, maintaining a manageable safety profile while achieving higher plasma steady-state concentrations of the active metabolite [137]. These studies laid the foundation for drug formulations and delivery methods for sphingolipid-type scaffold analogs.

### 5.4. α-Galactosylceramide

α-Galactosylceramide (α-GalCer), as shown in Figure 13, is a marine-based natural sphingolipid first isolated from the marine sponge *Agelas Mauritian* [138]. Based on the structural core, it is a glycosylated analog of ceramide. Due to its promising antitumor activity, numerous efforts have been made to synthesize this molecule to identify its therapeutic potential.

#### 5.4.1. Synthesis of α-Galactosylceramide

One of the most practical total syntheses of absolute anomeric confirmation of α-galactosylceramide was developed utilizing the chiral pool α-galactoside raffinose by Zhang et al. (Figure 6) [139].

Using the α-galactoside raffinose as the starting material provided the required α stereochemistry of the product. The α-linkage is vital for the biological functions of glycosphingolipids, and earlier attempts without α-linkage in the starting materials were challenging to synthesize [139]. The synthesis involved the protection of alcohols by benzylation to afford **6.1**. Acid-mediated selective cleavage of the β-fructofuranosidic linkage in **6.1** afforded hemiacetal (**6.2**). The subsequent reduction provided a diol (**6.3**). The primary alcohol in **6.3** was pivaloylated (**6.4**), and the secondary hydroxy was converted to azide (**6.5**). The pivaloylated alcohol on basic hydrolysis gave the primary alcohol (**6.6**), which was used as a handle to add the 12-carbon tail in the presence of NaH to afford (**6.7**). The azide group on (**6.7**) was reduced to amine, which on acylation afforded the amide (**6.8**). Finally, global debenzylation via catalytic hydrogenation over Pd/C afforded α-galactosyl ceramide, as shown in Figure 6 [139].

#### 5.4.2. Pharmacological Relevance and Mode of Action of α-Galactosyl Ceramide

α-GalCer’s mode of action relies on binding to the CD1 molecule on antigen-presenting cells, activating invariant natural killer T (iNKT) cells. These cells are a subset of immune cells that produce various cytokines, triggering T_h_1/T_h_2 immune responses. The cytokines effectively modulate the immune system to combat infections, autoimmune diseases, and cancer [140]. Various studies have demonstrated α-GalCer as a promising antitumor agent against multiple cancers, including melanoma, liver, and colon cancers [141].

Preclinical trial data provided mixed results, even though they demonstrated α-GalCer as well-tolerated, with prolonged overall survival [142]. However, efficacy was limited in humans due to challenges in effective delivery methods to enhance tumor-specific targeting and optimize immune cell interactions [138]. The full potential of α-GalCer can only be realized by developing novel target-specific delivery strategies with significantly improved clinical models to optimize new strategies [138].

### 5.5. Safingol

Safingol, also known as *L*-threo dihydrosphingosine, as shown in Figure 14, is a potential anticancer therapeutic agent [143]. It is a synthetic *L*-threo-stereoisomer of endogenous *D*-erythro-Sphinganine. Unlike typical sphingolipids with ‘*anti*-amino alcohol’ stereochemistry, it possesses atypical *syn* amino alcohol stereochemistry.

#### 5.5.1. Synthesis of Safingol

The total synthesis of safingol starts from _L_-Serine; the nucleophilic amine and hydroxy groups are protected to afford intermediate **7.2** (Figure 7). The allyl appendage for cross-metathesis was added by oxidation of the primary alcohol to an aldehyde, followed by chelation-controlled Grignard addition, resulting in cross-metathesis precursor (**7.3**). A long-chain alkene (**7.5**) is achieved using a cross-metathesis reaction with a Hoveyda–Grubbs catalyst. Finally, hydrogenation in the presence of Pd/C followed by Boc deprotection afforded safingol as an HCl salt [144].

#### 5.5.2. Pharmacological Relevance and Mode of Action of Safingol

Safingol demonstrates its promising anticancer properties as an inducer of autophagy and a modulator of multi-drug resistance. Safingol is a sphingosine kinase inhibitor (Ki −5 µM) and a milder sphingolipid protein kinase C (PKC) inhibitor (K_i_ −33 µM) [145]. Being a sphingosine kinase inhibitor, safingol shifts the sphingolipid rheostat towards ceramide accumulation, which is pro-apoptotic by reducing the S1P and thus limiting antiproliferative effects [146]. Even though safingol demonstrated significant in vitro anticancer activity, it showed limited in vivo activity. The most therapeutic advantage of safingol that is being explored is its ability to enhance the in vitro antitumor effect of various chemotherapeutic agents, such as cisplatin, mitomycin C, and doxorubicin, when used in combination. Safingol is the first sphingosine kinase inhibitor to enter clinical trials as an anticancer agent for various cancers, including colon and breast cancers, administered as a single agent with good, achievable plasma levels consistent with target inhibition and manageable hepatotoxicity. The results of preclinical trials also suggested that safingol is safe to co-administer with cisplatin, with dramatic potentiation in antitumor properties [145].

Shown in Figure 15 is sphingosine kinase I bound with a nanomolar inhibitor from Pfizer PF-543 [147]. To date, no binding studies have been reported on safingol’s interaction with sphingosine kinases I and II. Safingol has structural similarities with sphingosine and sphingosine-1-phosphate. We hypothesize that it may bind similarly with a polar head group in the vicinity of the catalytic cycle, as shown in Figure 15. Analogs of safingol docking studies in this crystal structure will guide the drug discovery efforts.

### 5.6. Spisulosine

Spisulosine (ES-285/1-deoxy-sphinganine), as shown in Figure 16, was first isolated from the clam *spisula polynyma*, showing its marine origin [148]. Due to its strong antiproliferative potential, it has gained notable importance. Some significant syntheses are discussed below.

#### 5.6.1. Synthesis of Spisulosine

One of the total syntheses of Spisulosine by Fabišíková, M. et al. (Figure 8) involved a substrate-controlled aza-Claisen rearrangement, which set the *anti*-amino alcohol motif. The methyl head group was created by deoxygenation. The 15-carbon chain was obtained by Wittig olefination and subsequent hydrogenation, as shown in Figure 8 [148].

As shown in Figure 8, D-isoascorbic acid **8.1** was converted to the ester (**8.2**). Benzylation of **8.2** followed by reduction afforded the acetonide alcohol (**8.4**). Subsequent oxidation and witting reactions provided the olefinic ethyl ester (**8.5**). Mesylation followed by S_N_2 addition afforded a mixture of enantiomers (**8.7**) and (**8.8**). Several method development studies were conducted to enhance the diastereoselectivity of the *anti*-**8.7** over the *syn*-**8.8**. After optimization, starting from **8.7**, a carbamate intermediate (**8.9**) was synthesized upon mesiyllnitrile oxide (MNO) treatment. The olefin in (**8.9**) was subjected to ozonolysis and reduction to afford alcohol (**8.10**). The alcohol was converted to an iodo intermediate followed by reduction, which afforded (**8.12**). *N*-benzylation followed by acetonide deprotection afforded Vic-diol (**8.14**). Periodate oxidation of vicinal diol, which was then treated with a non-stabilized ylide, produced a barely separable mixture of olefinic intermediates (**8.15**). Repeated chromatographic separation afforded pure (*Z*)-**8.15**. Saturation of the double bond and removal of both benzyl ether protecting groups under catalytic hydrogenation gratifyingly provided spisulosine.

An improved and shorter synthetic method by Amarante, G.W. et al. [149] (Figure 9), discussed here involves a Morita–Baylis–Hillman adduct. Hexadecanal (**9.1**) was subjected to a Morita–Baylis–Hillman reaction in the presence of DABCO with methyl acrylate, resulting in the conversion of the aldehyde to a methyl ester (**9.2**). The yield of this first step was not satisfactory, as hexadecanal, a long-chain aldehyde, is not a desirable substrate for the Morita–Baylis–Hillman reaction. The subsequent hydrolysis of ester to acid (**9.3**) was followed by reductive amination, which afforded the benzylamine (**9.5**), and finally, global deprotection afforded free amino alcohol spisulosine (Figure 9) [149].

#### 5.6.2. Pharmacological Relevance and Mode of Action of Spisulosine

On studying the sequence of molecular level events triggering apoptosis and examining its effects on the “cell death markers”, it was observed that spisulosine triggers apoptosis by activating caspases 3 and 12 and modifying the phosphorylation of P53. Spisulosine induces phosphorylation of MARCKS just like the sphingonine-related lipids. However, spisulosine is not an inhibitor of PKC; instead, it is an activator of PKC at the cellular level, unlike its analogous sphingonine-related lipids. Spisulosine did not affect other pathways involved in cell survival/apoptosis, such as JNK, Erks, or Akt, suggesting the triggering of an atypical cell death [150]. In in vitro studies, spisulosine showed an activity of 1–10 μM against both PC-3 and LNCaP prostate cancer cell lines by increasing intracellular ceramide levels. This ceramide level increase was not observed when challenged with ceramide synthase inhibitor fumonisin B1. Ceramide synthase is involved in the salvage pathway towards the biosynthesis of ceramide. This indicates that cellular ceramide levels increase is via a *de novo* biosynthetic pathway and shows how the sphingolipid’s complex biochemistry is regulated and compartmentalized [151].

Activation of PKCζ, a target protein of ceramide, was also observed in PC-3 and LNCaP prostate cancer cell lines. The results obtained with specific inhibitors of various pathways inferred that the antiproliferative effect induced by spisulosine in prostate cancer cells was independent of peroxisome proliferator-activated receptor gamma (PPARγ), p38/classical protein kinase C (PKCs) pathways, Jun *N*-terminal kinase (JNK), and phosphatidylinositol 3-kinase/(PI3K/Akt). Another study indicates that spisulosine cytotoxicity is due to the prevention of the formation of stress fibers, which ultimately decreases the activity of Rho proteins [152].

Clinical trials were conducted on spisulosine owing to its antiproliferative activity. The results were not promising; one of the phase 1 trials indicated induction and elevation of the liver enzymes, resulting in dose-limiting for spisulosine and low antitumor activity [153]. Other clinical trials also indicated hepato- and neurotoxicity as schedule-independent dose-limiting adverse events [2].

Another phase 1 clinical trial was designed to identify the recommended and maximum tolerated doses (MTD) for phase (II) trials. This study also evaluated the safety profile, pharmacokinetics, and preliminary efficacy data in patients with advanced solid tumors. The results indicated dose level VIII (200 mg/m^2^) as the MTD, and dose level IX (160 mg/m^2^) was defined as the RD. This study also indicated limited antitumor activity [154].

These adverse events correlate to the poor physicochemical properties of spisulosine resulting from the structure, including the long hydrocarbon chain’s high lipophilicity and intramolecular hydrogen-bond formation between the hydroxyl and the amine functionality, hindering the ionization of the functional groups and lowering the pH to much lower values than commonly seen for primary amines. The basic amine salts improve the water solubility. Even though adequate solubility in salt form may be achieved, another issue is the tendency towards gel formation due to aggregation of the lipophilic portion of the molecule over time in plasma. These features interfere with the infusion of aqueous solution. It was evidenced in the clinical trial that the time for infusion increased up to 24 and up to 72 h, which would result in gel formation. The Drug-related adverse reactions during the clinical trials included nausea, pyrexia, injection site reactions, vomiting, and one case of death of the patient due to neuro- and hepatotoxicity [2]. This observation led to halting clinical trials for spisulosine.

These studies warrant the development of new drug delivery platforms for the therapeutic applications of spisulosine and its analogs.

### 5.7. Jaspine B

#### 5.7.1. Synthesis of Jaspine B

Jaspine B (pachastrissamine), as shown in Figure 17, is an anhydrophytosphingosine extracted from the marine sponge *jaspis* sp. [155]. It has effective anticancer activity against several human carcinomas [156]. The gram-scale synthesis of jaspine B starts from naturally abundant _L_-Serine utilizing a chiral pool strategy to afford an advanced chiral intermediate, Dutta lactone (**10.6**, Figure 10). The total synthesis design strategy involved using a chromophoric Cbz-_L_-Ser over *N*-Boc-_L_-Ser functionality. This facilitated the robust purification of several intermediates using the Combi-Flash purification system for scale-up, affording an advanced chiral intermediate, Dutta lactone. Jaspine B, as such, is a non-UV active natural product on thin-layer chromatography. The strategic incorporation of Cbz-protecting functionality in the first step significantly aided in synthesizing multi-gram-scale batches and facilitated chromatography. It is very well tolerated in acidic and basic environments and was retained until the precursor step of jaspine B, starting from _L_-Ser.

The synthesis of jaspine B employs a chiral pool strategy, starting from _L_-Ser (Figure 10). The initial steps involve *N*-Cbz protection, followed by amidation of the carboxylic acid and acetonide protection, to afford the Weinreb amide (**10.1**). This scale-up is a very robust, efficient methodology executed on a 50 g scale. Weinreb amide (**10.1**) is subjected to Grignard addition using allyl Mg. bromide followed by double bond migration to afford the thermodynamically favored unsaturated trans ketone (**10.3**). A chelation-controlled reduction of a trans ketone using a freshly prepared zinc borohydride reagent afforded the trans-alcohol intermediate (**10.4**). Acroylation of **10.4** provides the acrylate intermediate (**10.5**) as a greasy liquid. Utilizing Grubbs Ist Gen. catalyst, a ring-closing metathesis (RCM) of **10.5** acrylate affords Dutta lactone (**10.6**, Figure 10). Utilizing the Teledyne Combliflash chromatographic system, all intermediates can be subjected to chromatographic separation, resulting in reduced solvent utilization and increased efficiency by reducing time and manpower.

Dutta lactone is a second-generation advanced chiral intermediate with inherent chirality, unsaturated lactone (Michael acceptor), and protected nucleophiles (OH, NH_2_). Acetonide deprotection of this lactone yields the thermodynamically favorable enantiopure bicyclic furafuranone as the sole product. The required all-syn tri-substitutions are achieved in this transformation. Functional group transformation of the lactone in bicyclic furafuranone provides an *N, O*-protected jaspine B precursor (**10.11**). Global deprotection affords jaspine B as HCl salt [157], as shown in Figure 10.

#### 5.7.2. Pharmacological Relevance and Mode of Action of Jaspine B

Structurally, jaspine B resembles ceramide (Figure 17). This close similarity makes it a probe and interferes with ceramide metabolizing enzymes. Jaspine B was extensively tested in several in vitro and in vivo studies. Some mechanistic studies involved necroptosis, which is followed by cell death [158] and mitosis-mediated programed cell death, among others. Datta et al. developed an efficient synthetic route using an _L_-Serine-derived bicyclic lactone as an advanced chiral building block [159]. Utilizing this synthesis, our groups have accomplished the synthesis of a gram quantity of jaspine B. We have used this knowledge to understand membrane binding potential and develop a sphingolipid-based structural core formulation for novel drug delivery platforms.

In collaborative efforts, a novel liposome drug delivery system was developed using jaspine B to address its low bioavailability issues, with a control, and to improve its therapeutic efficacy [160,161] There have also been several pharmacokinetics and ADME studies aimed at understanding the mechanisms of intestinal absorption, distribution, metabolism, and excretion of jaspine B in rats [162]. Jaspine B has poor bioavailability, with only 6.2% [162] absorption. Several studies have suggested an increase in the bioavailability of jaspine B through co-administration with bile salts and micelle formation. In this formulation, the lipophilic cholesterol portion may interact with the lipophilic portion of jaspine B. Jaspine B is noted to be a highly tissue-distributed compound, with higher concentrations found in the brain, kidney, heart, and spleen [162].

To gain a deeper understanding of the binding interactions of jaspine B and its analogs, a molecular modeling study utilizing a monomeric hSMSr [PDB: 8W9Y] (Figure 18) will provide valuable insights into its interactions with these ligands.

Recently, cryo-electron microscopic structures of human SMSr in complexes with ceramide, diacylglycerol/phosphoethanolamine, and ceramide/phosphoethanolamine (CPE) were reported [163]. Interestingly, it is a hexameric arrangement with catalysis located between the transmembrane helices. The authors have identified a catalytic pentad E-H/D-H-D at the interface between the lipophilic and hydrophilic sequences of the catalytic pentad. The mechanism involves PE-PLC (phosphatidylethanolamine-phospholipase C) hydrolysis, followed by the transfer of the phosphoethanolamine motif to ceramide. To our knowledge, this is the first time mechanistic evidence with a crystal structure has been identified. These studies suggest that structural similarities with ceramide have the potential to bind in the same environment as the enzyme substrate. This further substantiates studies on pharmacological interventions involving jaspine B and ceramide, contributing to drug discovery initiatives.

### 5.8. d-Erythro-MAPP(D-e-MAPP)

#### 5.8.1. Synthesis of d-Erythro-MAPP(D-e-MAPP)

D-e-MAPP, as shown in Figure 19, is a stereochemically opposite enantiomer to the typical *L*-sphingolipids, and only the D-stereoisomer has exhibited the antiproliferative effect. The *L*-stereoisomer (*L*-e-MAPP) is inactive [164]. Therefore, stereospecific syntheses were necessary to afford D-enantiomer selectively.

One of the syntheses by Chang, Y.-T. et al. [165] involve a single-step coupling reaction between (1*S*,2*R*)-2-amino-1-phenylpropan-1-ol (**11.2**) and tetradecanoyl chloride (**11.1**) in the presence of the organic base pyridine and THF as the solvent, to afford d-erythro-MAPP (Figure 11) [165].

#### 5.8.2. Pharmacological Significance of d-Erythro-MAPP

Various studies have demonstrated that d-e-MAPP is an inhibitor of alkaline ceramidase both in vitro and in cells [166,167]. It elevates endogenous ceramide levels with growth suppression and cell cycle arrest effects. D-e-MAPP has been used as an alkaline ceramidase inhibitor to perform detailed studies demonstrating the role of ceramidases in regulating the endogenous levels of ceramide [164,166,167]. D-e-MAPP showed no interesting effects on other ceramide metabolism enzymes, including sphingomyelinase and glucocerebroside synthase. It is also ineffective against neutral and acidic ceramidases [168].

### 5.9. B13

#### 5.9.1. Synthesis of B13

B13, as shown in Figure 20, is a small molecule that targets ceramidases. It is synthesized by Bai, A. et al. (Figure 12) by a simple amidation of tetradecanoic acid (**12.2**) and 2-amino-1-(4-nitrophenyl)propane-1,3-diol (**12.1**). B13 is a potent acid ceramidase inhibitor, with a primary target being the lysosomal acid ceramidase enzyme. This selectively inhibits the acidic form of acid ceramidase. B13 has also shown a minor effect on neutral ceramidase. B13 is being explored as an antiproliferative agent because it enhances ceramide levels in cancer cells [169].

#### 5.9.2. Pharmacological Significance of B13

B13 prevents the aminolysis of ceramide to form sphingosine and a fatty acid by binding to the acid ceramidase enzyme’s active site, resulting in ceramide-induced apoptosis [170]. The close structural similarity with the substrate Cer makes it a suitable inhibitor targeting ceramidase. B13 was found to be highly potent in vitro against various cancer cell lines; however, it faced challenges in reaching the lysosomal compartment within the cells, potentially limiting its in vivo efficacy. A lysosomal formulation of B13 utilizing an *N*, *N*-dimethylglycine (DMG) ester-modified prodrug (DMG-B13 prodrug) was synthesized to address issues related to drug delivery. The DMG-B13 formulation facilitated its accumulation in acidic lysosomes due to the basicity of the ionizable amine, resulting in a markedly improved cellular action [169]. Notably, a one-step amidation can provide a pharmacological agent to perform pharmacological studies targeting ceramidase. A quick literature search for the crystal structure of ceramidase in the Protein Data Bank (www.rcsb.org) resulted in 27 deposits from several species. Shown in Figure 21 is the crystal structure of human acid ceramidase in a covalent complex with the small molecule carmofur [109]. A molecular modeling study with B13 in this enzyme depicts the crucial interactions necessary for enzyme inhibition. Finally, the bioactive sphingolipid-based medicinal agents, their mechanism of action, and their developmental status have been summarized and are presented in Table 3. 

## 6. Conclusions

Sphingolipids have a polar head group and a lipophilic tail that anchors them to the membrane, which is critical for their biological activity, for maintaining membrane structural integrity, and for regulating cellular signaling. In both preclinical and clinical settings, numerous small molecules and natural products that mimic the sphingolipid structural core have emerged as promising tools for modulating key enzymes involved in sphingolipid biosynthesis and metabolism. This review focuses on the utility of the chiral pool strategy, particularly the use of amino acids and carbohydrate scaffolds, C-C bond formation, stereoselective syntheses, and functional group transformations, as a foundational approach in synthesizing such analogs.

Sphingolipid biosynthesis in mammalian cells typically initiates with _L_-Serine, leading to the formation of oxysphingolipids. However, recent studies reveal that serine palmitoyltransferase (SPT), the rate-limiting enzyme in this pathway, also catalyzes reactions involving other amino acids, particularly _L_-Alanine, generating deoxysphingolipids. Both types of sphingolipid classes can enter the sphingolipid biosynthetic flux, and this disruption in the normal lipid metabolism leads to various pathological conditions.

Additionally, this review highlights the significance of spatial compartmentalization and tissue-specific expression of enzymes in regulating sphingolipid metabolic pathways and their role in disease manifestation. Integrating transcriptomic and bioinformatic approaches focused on sphingolipid-associated enzymes can significantly advance early-phase drug discovery and structure–activity relationship (SAR) exploration.

Despite these advances, a notable gap remains in the application of bioconjugation chemistry to track sphingolipid trafficking and intracellular localization. While this review does not extensively cover such strategies, fluorescent NBD-sphingolipid derivatives have proven invaluable in visualizing these pathways and hold promise for refining enzyme-targeted therapeutic interventions.

Formulation advancements, such as the development of the orally disintegrating Cycle Vita™ (fingolimod) and nanoliposomal fenretinide, demonstrate the critical role of drug delivery systems in translating sphingolipid-based molecules into viable therapeutics.

Therefore, by carefully integrating molecular design and computational tools, future research on drug delivery methodologies can leverage the enigmatic nature of sphingolipid metabolism and signaling to unlock novel therapeutic avenues.

## Data Availability

Not applicable.

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
