# Peer review of "A Review Discussing Synthesis and Translational Studies of Medicinal Agents Targeting Sphingolipid Pathways"

_biomolecules, 2025, doi:10.3390/biom15071022_

Round 1
Reviewer 1 Report (Previous Reviewer 1)
Comments and Suggestions for Authors
The review “Synthesis, molecular modeling, and translational studies of medicinal agents targeting sphingolipid pathways” is devoted to the syntheses and drug development efforts of sphingolipid-based medicinal agents (Myriocin, Fingolimod, Fenretinide, Safingol, Spisulosine (ES-285), jaspine B, D-e-MAPP, B13, 23 and α-Galactosylceramide) of preclinical and clinical studies. The text of the review is well structured and understandable. The review is supplemented with original illustrations that improve the perception of the text. However, the review contains some shortcomings, which are pointed out separately below. The main comments concern the lack of a detailed description of the methodology, as well as the insufficient elaboration of the figure captions and designations in the figures.
In my opinion, the review has been significantly improved after resubmitting. It may be published in a journal.
I have one comment:
Figure 4. A PubMed search of “Sphingolipid” and “Disease States” generated approximately 3000 research reports with increased efforts during the past two decades.
The y-axis values suggest that there were more than 3000 research reports.
Author Response
The review “Synthesis, molecular modeling, and translational studies of medicinal agents targeting sphingolipid pathways” is devoted to the syntheses and drug development efforts of sphingolipid-based medicinal agents (Myriocin, Fingolimod, Fenretinide, Safingol, Spisulosine (ES-285), jaspine B, D-e-MAPP, B13, 23 and α-Galactosylceramide) of preclinical and clinical studies. The text of the review is well structured and understandable. The review is supplemented with original illustrations that improve the perception of the text. However, the review contains some shortcomings, which are pointed out separately below. The main comments concern the lack of a detailed description of the methodology, as well as the insufficient elaboration of the figure captions and designations in the figures.
Response:
Thank you for your time and comments. The figure indents were revisited and edited with details on the mechanistic details. The context is edited to support the big picture of drug discovery efforts in the small molecules which undergone preclinical and clinical studies. We have reviewed several articles and noticed either the synthesis discussion or modeling or invitro studies are not located in one scientific report. We focused on this main theme and discussed in this review.
In my opinion, the review has been significantly improved after resubmitting. It may be published in a journal.
I have one comment:
Figure 4. A PubMed search of “Sphingolipid” and “Disease States” generated approximately 3000 research reports with increased efforts during the past two decades.
The y-axis values suggest that there were more than 3000 research reports.
Response: : It is more than “30,000” we have edited this finding.
Reviewer 2 Report (Previous Reviewer 2)
Comments and Suggestions for Authors
I appreciate authors efforts in revising the manuscript extensively. Authors have addressed all the raised concerns, and now the manuscript can be accepted in its current form.
Author Response
No issues – Good Job
Thank you for the suggestions during this entire process. We are very encouraged.
Reviewer 3 Report (Previous Reviewer 3)
Comments and Suggestions for Authors
In my assessment, the manuscript submitted by Mateen et al. is not yet suitable for publication in Biomolecules, despite the authors' efforts to revise and improve the work.
Firstly, I would like to highlight that Biomolecules is a journal that primarily publishes work by experienced and well-established researchers. In its current form, this manuscript appears more as a training exercise for early-career scientists than a contribution that meets the journal's scientific and editorial standards.
A significant concern remains the confusion between the formats of an Original Research Article and a Review. Upon reevaluating the manuscript, it is clear that this distinction is still not properly understood by the authors. For their reference, I have attached a summary outlining the key differences between these two formats. This issue must be resolved before any further consideration for publication.
Specifically, all molecular modeling data presented in the manuscript-particularly in Figures 9, 15, 18, 20, and 22-should be removed, as these appear to represent unpublished original research, which is not appropriate in a review article.
While the use of figures is appreciated, they should be constructed using experimentally determined structures available in databases such as the RCSB Protein Data Bank. Additionally, all synthetic schemes and chemical structures (presumably created with ChemDraw) must be provided in high resolution, as they currently appear blurry and unprofessional in the manuscript.
Finally, the title of the manuscript is misleading, as it implies the work is an original research article. To avoid confusion, the title must be revised to clearly indicate that the manuscript is a Review.
In summary, the manuscript could only be reconsidered once these fundamental issues-particularly the inappropriate inclusion of unpublished data and the lack of clarity about the article type-are fully addressed.

Author Response
Comments: In my assessment, the manuscript submitted by Mateen et al. is not yet suitable for publication in Biomolecules, despite the authors' efforts to revise and improve the work.
Firstly, I would like to highlight that Biomolecules is a journal that primarily publishes work by experienced and well-established researchers. In its current form, this manuscript appears more as a training exercise for early-career scientists than a contribution that meets the journal's scientific and editorial standards.
A significant concern remains the confusion between the formats of an Original Research Article and a Review. Upon reevaluating the manuscript, it is clear that this distinction is still not properly understood by the authors. For their reference, I have attached a summary outlining the key differences between these two formats. This issue must be resolved before any further consideration for publication.
Specifically, all molecular modeling data presented in the manuscript-particularly in Figures 9, 15, 18, 20, and 22-should be removed, as these appear to represent unpublished original research, which is not appropriate in a review article.
While the use of figures is appreciated, they should be constructed using experimentally determined structures available in databases such as the RCSB Protein Data Bank. Additionally, all synthetic schemes and chemical structures (presumably created with ChemDraw) must be provided in high resolution, as they currently appear blurry and unprofessional in the manuscript.
Finally, the title of the manuscript is misleading, as it implies the work is an original research article. To avoid confusion, the title must be revised to clearly indicate that the manuscript is a Review.
In summary, the manuscript could only be reconsidered once these fundamental issues-particularly the inappropriate inclusion of unpublished data and the lack of clarity about the article type-are fully addressed.
Response: We sincerely appreciate the reviewer comments and took it very seriously and edited in the past. The finding of molecular modeling of these preclinical and clinical studies is very prominent. With the recent advancements in molecular modeling applications, using these early on would help to design better hits. We have removed these findings and agree, as these are novel studies should be published separately. But, we did include the enzyme structures only with PDB files and provided the details of value of molecular modeling using those PDB files to the readers. We strongly believe, providing the crystal structures source will help the drug discovery team ahead.
The title is also edited to reflect the philosophy of “Review” rather than an original article. The chemdraw figures were redrawn and pasted as JPEG format to improve the resolution. We can surrender the .CDX files to MDPI if need be.
This manuscript is a resubmission of an earlier submission. The following is a list of the peer review reports and author responses from that submission.
Round 1
Reviewer 1 Report
Comments and Suggestions for Authors
The review “Synthesis, molecular modeling, and translational studies of medicinal agents targeting sphingolipid biochemistry” is devoted to the syntheses and drug development efforts of sphingolipid-based medicinal agents (Myriocin, Fingolimod, Fenretinide, Safingol, Spisulosine (ES-285), jaspine B, D-e-MAPP, B13, 23 and α-Galactosylceramide) of preclinical and clinical studies. The text of the review is well structured and understandable. The review is supplemented with original illustrations that improve the perception of the text. However, the review contains some shortcomings, which are pointed out separately below. The main comments concern the lack of a detailed description of the methodology, as well as the insufficient elaboration of the figure captions and designations in the figures. I believe that it can be published after major revision.
Main issues:
- Lines 49-51. The biosynthesis starts 49 from the cellular chiral pool, L-Ser, resulting in a family of Oxysphingolipids. while L-Ala 50 results in a family of deoxysphingolipids [8] (Figure 1).
It would be desirable to add information about the specified precursors to Figure 1.
- Figure 1.
1-Deoxysphingosine is shown twice with different structural formulas. Please check the formulas and names of the compounds.
- Figure 2.
Please add an explanation of the letters R, s, S in the figure.
In the de novo pathway on the diagram, L-serine should be specified.
- Figure 3.
The letters and numbers in the picture are not visible. Please make them larger and clearer.
- 5. Biological significance of Sphingolipids and derivatized sphingolipids
The information provided in this section is insufficient. Tables should be included in the section indicating the main sphingolipids and their biomedical significance.
- Lines 232-239
The methodology is described very sparingly. The authors are advised to create a Materials and methods section where the procedures are described in detail. This is necessary so that other researchers can repeat the experiments.
- Authors are also encouraged to create an abbreviations section.
- Figure 8
Please explain in more detail what is shown in the picture.
- Lines 353-354. However, unlike several retinoic acid analogs, it did not exhibit dermatological intervention.
Please clarify what interventions are meant.
- Discussion
In my opinion, the discussion section needs to be revised and expanded. Or it should be rewritten into a conclusions section. The meaning of compartmentalization mentioned in this section should be highlighted in the text of the article.
- enantioselective syntheses
As for the enantioselective syntheses indicated in the abstract, its necessity for each compound should be more clearly indicated.
Reviewer 2 Report
Comments and Suggestions for Authors
Manuscript by Srinath et al., presents extensive coverage of sphingolipid-based agents, including synthesis, molecular modelling, and pharmacological relevance. However, there are some strengths where I appreciate the authors, but there rae some points, where I would like to suggest authors to revise the manuscript for maintaining the flow and depth of the review and correcting the technical issues.
- Firstly, what I have noticed in the review is that you have referred lot of docking studies. But in my point of view, docking is meaningless until it is validated. Docking alone can not produce any authentic and validated results. It will be better if authors confirm that all these studies which authirs have referred are validated either by lab studies, or atleast by molecular dynamics to confirm the binding stability. If you have referred any study which is docking alone, it should be removed from the review. Without validation atleast by molecular dynamics simulation, the prediction alone with docking has limited scope and applications.
- Your figure 4 is not correct. You have mentioned 3000 papers total. But what each bar is representing? Where is the x and y axis legend? Please see the figure 2 of this paper (https://pubs.rsc.org/en/content/articlelanding/2025/tb/d4tb02068e), and follow the pattern of presenting this figure of number of papers. Further, present the insights of this figure. What these 3000 researches means and what are they highlighting? It could be more meaningful with interpretation of this figure in text as well.
- Table 1. You need to add separate row and column for references. Do not add references after the small molecule. Write every molecule individually and then place the right reference in separate column.
- There should not be a heading “discussion” in a review article. It should be conclusion. Please recheck.
- Further, you have discussed many compounds, but the comparative analysis or critical evaluation of efficacy, toxicity, or clinical relevance is lacking in the manuscript.
- Title of the manuscript is not right. Are you targeting sphingolipid biochemistry? Or you are presenting bioactive sphingolipids exhibiting numerous pharmacological effects targeting multiple diseases/receptors/pathways etc.? Please correct the title.
- I suggest to add one comparative table summarizing each compound’s pharmacological potential, limitations, and current trial status in clinical settings. I can see that you have not written anything about the future perspectives and limitations on bioactive sphingolipids and their potential medical use for targeted therapy in clinical settings. You must add a table, as well as a detailed paragraph on drug delivery challenges and strategies to strengthen the translational perspective.
- Though the topic is timely and conceptually rich, what is the new frameworks or hypotheses you wanna present. Please add in abstract as well as in your concluding remarks.
- Some paragraphs are too long and overloaded with details, affecting the readability. Please rephrase those paragraphs and better to split in multiple sentences.
- You have to elaborate Sphingolipidome for readers understanding.
- Line 50…..Oxysphingolipids. while……period before while is not right.
- Some inconsistency with figure references here. The text mentions key bioactive sphingolipids (Cer, GCer, C1P, Sph, S1P)“depicted in Figure 5” (line 59-60) but then refers to “Figure 1” in line 61. Later, line 75-76 is the caption for Figure 1 (deoxy- and oxy-sphingolipids).
- The concept of lipid rafts is reintroduced, noting their presence in membranes and even viral envelopes, and then immediately tying their significance to theCer/S1P rheostat. The connection between lipid rafts and the rheostat could be explained a bit more.
- Table 1 should be presented after section 5. Not placed correctly.
- Everywhere you have repeatedly used“P13K” instead of “PI3K”. Please correct it. Similarly, “p7OS6K” should be “p70-S6K” (p70 S6 kinase)​.
- Line 370: Are you sure about fenretinide’s efficacy in Phase III? Is it done and approved? Please recheck.
- Line 521-525: I don’t think this is the correct reference here. Moreover, you are talking about several clinical trials, and mentioning only one reference. Not correct
Reviewer 3 Report
Comments and Suggestions for Authors
The manuscript “Synthesis, molecular modeling, and translational studies of medicinal agents targeting sphingolipid biochemistry” of Mateen et al. aims to describe enantioselective syntheses and drug development efforts of sphingolipid-based medicinal agents of preclinical and clinical stages. Despite the efforts of the authors the manuscript is only a preliminary draft. Many statements are not accurate from a scientific point of view and overall, the review appears unfocused and fails to align with its intended objectives. The following points highlight the main issues that render it unsuitable for publication."
1) The major issue is the presence of molecular modeling experiments conducted by the authors as reported at line 218 “We have also performed novel molecular modeling studies” whereas in a review, all the progress in the field should be covered and discussed without experimental contributions."
2) The manuscript's title gives the impression of an original research paper, which does not align with its actual content as a review. "A more suitable title might be: 'Medicinal Agents Targeting Sphingolipid Biochemistry: A Systematic Review.'
3) "I suggested that the authors revise the manuscript by providing a comprehensive discussion solely on the aspects related to 'Medicinal Agents Targeting Sphingolipid Biochemistry' as found in the most recent literature, limiting the scope to studies published in the last 10 years at most.
4) "Paragraph 2, starting at line 78, along with subsections 2.1, 2.2, and 2.3, should be simplified and condensed."
5) "At line 117, the model by Campopiano et al. is mentioned as the twenty-fourth citation, but it is missing from the reference list."
7) The discussion section at line 679 is inconsistent and lacks critical elements that highlight the strengths and weaknesses of the current scientific landscape in the field of sphingolipid biochemistry.
6) Finally I suggest that the authors conduct a careful and critical revision of the manuscript, significantly improving also the quality of the English language."
Comments on the Quality of English Language"The quality of the English language is quite poor and basic, with many repetitions between nearby sentences."